# Multimodal AI and tumour microenvironment integration predicts metastasis in cutaneous melanoma

Tom W. Andrew [1] ✉, Marc Combalia[2], Carlos Hernandez[2], Sydney Grant [2], Gyorgy Paragh [3], Susanna Puig [4], Grant Mc Arthur [5], Grant Richardson[1,2], Phil Sloan[1,2], Sophia Z. Shalhout [6,7], Ruth Plummer [1,8] & Penny E. Lovat[1,2]

Accurate prognostication is essential to guide clinical management in localised cutaneous melanoma (CM), the form of skin cancer with the highest mortality. While the tumour microenvironment (TME) plays a key role in disease progression, current staging systems rely on limited tumour features and exclude key clinicopathological prognostic features. Here we show that MelanoMAP, a multimodal AI model integrating TME-derived digital biomarkers and clinicopathological features from over 3,500 histology slides, improves prognostication of localised CM. MelanoMAP achieved a C-index of 0.82, a 24% improvement over traditional AJCC staging (0.66) and consistently outperformed clinicopathological-only models across six international patient cohorts. SHAP analysis identified TME-derived digital biomarkers, alongside traditional clinicopathological factors including age, mitotic count, and Breslow depth, were critical determinants of metastatic risk. MelanoMAP establishes a potential foundation for precision oncology in CM, demonstrating how AI-driven digital biomarkers can advance personalised prognostication and inform clinical-decision making.

Cutaneous melanoma (CM) is the leading cause of skin cancer mortality[1,2]. Most melanoma-related deaths arise from patients initially diagnosed with localised disease[3]. Effective adjuvant immunotherapy treatments are available for high-risk localised CM but carry considerable morbidity risk, underscoring the need for more precise risk stratification to predict metastatic progression to optimise follow-up and treatment strategies for patients with localised CM.

Current prognostication for localised CM relies on the American Joint Committee on Cancer (AJCC) 8th edition staging system, which stratifies stage I/II tumours based on Breslow depth and ulceration[4]. While these factors are important, the system currently excludes key clinicopathological prognostic factors, including mitotic count which was removed from the latest edition[4], and fails to account for features in the tumour microenvironment (TME), limiting its ability to predict disease progression. The TME plays a pivotal role in CM biology, driving tumour growth, immune evasion and metastasis through complex interactions[5–7]. Tumours can release growth factors, cytokines, and extracellular vesicles that enable early locoregional and distant spread[8–10]. These molecular signals continue to influence clinical outcomes even after complete surgical excision of the primary tumour[8–10]. These processes highlight the TME's dynamic role in shaping disease behaviour and its potential as a rich source of

[1]Translation and Clinical Research Institute, Newcastle University, Newcastle Upon Tyne, UK. [2]AMLo Biosciences, Newcastle Upon Tyne, UK. [3]Department of Dermatology, Roswell Park Comprehensive Cancer Center, Buffalo, NY, USA. [4]Dermatology Department, Hospital Clinic of Barcelona, University of Barcelona, IDIBAPS & CIBERER, Instituto de Salud Carlos III, Barcelona, Spain. [5]Sir Peter MacCallum Department of Oncology, University of Melbourne, Melbourne, Victoria, Australia. [6]Mike Toth Head and Neck Cancer Research Center, Division of Surgical Oncology, Department of Otolaryngology-Head and Neck Surgery, Mass Eye and Ear, Boston, MA, USA. [7]Department of Otolaryngology-Head and Neck Surgery, Harvard Medical School, Boston, MA, USA. [8]Sir Bobby Robson Cancer Trials Research Centre, Newcastle Hospitals NHS Foundation Trust, Newcastle upon Tyne, UK. ✉e-mail: tom.andrew@nhs.net

prognostic information. However, the vast number of features within the TME and their intricate spatial relationships make comprehensive assessment challenging with conventional histopathological methods, necessitating AI-driven approaches to uncover previously unidentified prognostic patterns.

Advances in computer vision and deep learning have transformed histological cancer research, enabling the identification, analysis, and interpretation of digital biomarkers[11]. When integrated with clinicopathological data, these image-derived features can form multimodal AI models that enhance risk stratification and prognostication[12,13]. Building on this foundation, the study combines deep learning-based histology image features with clinicopathological features from six international CM cohorts to develop a multimodal AI model for localised melanoma prognostication. By focusing on uncovering digital biomarkers within the TME and combining these with known clinicopathological prognostic features, this approach not only refines prognostic insights but also suggests avenues for future innovations in digital biomarker discovery and AI clinical prognostication.

## Results

### Cohort assembly and baseline characteristics

Whole slide images (WSI) from five international cohorts ($n = 3236$) of localised CM were available for the study. Of these, 23 WSI had missing clinical information, and 33 had WSI of insufficient quality/completeness for analysis and were excluded. These cohorts were combined into a multinational database ($n = 3180$) before being split into training and test cohorts (80:20) for model development. An additional 481 WSI were included from a sixth cohort for external testing. Of these, WSI from 4 patients were of insufficient quality/completeness for analysis which were excluded. The data assembly process is outlined in Fig. 1.

Baseline features of the training, validation and external test cohorts are presented in Table 1 and institutional breakdown are presented in Supplementary Table 1. The median age of patients was comparable across cohorts (58 vs 62 vs 62 years, $p = 0.079$) for the training, test, and external test cohorts, respectively. There was no significant difference between sex across cohorts, 47% male in the training, 50% in the test, and 47% in the external test cohort ($p = 0.71$). This was also true for Breslow depth ($p = 0.096$) and mitotic count ($p = 0.66$). 10% of the training, 10% of the test, and 16% of the external UK test cohort had distant metastasis ($p = 0.083$), occurring over a comparable median time to metastasis of 83–87 months across all groups ($p = 0.022$). Despite the similarities across the groups, differences were detected in the distribution of tumour anatomical site and histological subtypes in the external test cohort compared to other groups ($p < 0.05$). Greater differences in patient characteristics were detected in the institutional breakdown, reflecting greater institutional specific variability and granular heterogeneity across centres.

### Development and validation of the MelanoMAP model

In this study, Melanoma Multimodal AI Prognostication (MelanoMAP) was developed as a multimodal AI pipeline consisting of a histopathological image and tabular clinicopathological features analysis. The imaging analysis pipeline processes WSIs stained with H&E, as well as by immunohistochemistry (IHC) for the expression of Activating Molecule in Beclin-1 Regulated Autophagy 1 (AMBRA1) and Loricrin, key to tumour suppression and epidermal terminal differentiation, respectively[14–16]. The stains were obtained from adjacent anatomical slides within the same primary tumour. The inclusion of IHC stains enhances the model's ability to detect microenvironmental changes by capturing autophagy and cellular differentiation regulatory mechanisms in the epidermis driven by TME interactions, supporting biology-guided model explainability[10]. A modified U-Net architecture was employed for segmentation. WSIs were pre-processed to distinguish tissue regions (foreground) from the background and segmented into smaller patches for analysis. The U-Net model generated segmentation masks categorising regions into background, epidermis, tumour, and IHC staining. The segmentation performance was robust across all stain types, with F1-scores of 0.92 for H&E, 0.95 for AMBRA1, and 0.94 for Loricrin, and Intersection over Union (IoU) scores exceeding 0.90 for each stain type. These results confirmed the model's accuracy in delineating key histological structures.

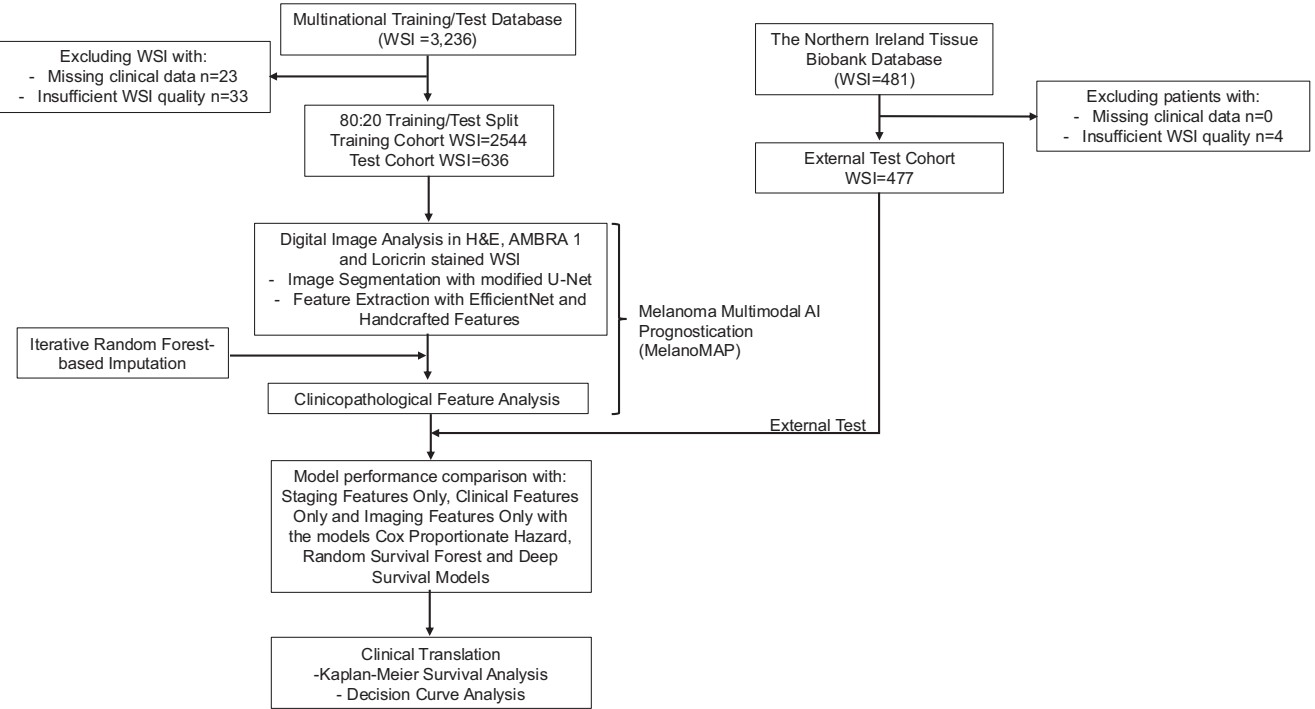

**Fig. 1 | Study design and workflow of patient selection.** This diagram outlines the data processing pipeline for MelanoMAP, a multimodal AI prognostic model designed to predict time-to-recurrence in localised (AJCC stage I/II) cutaneous melanoma (CM).

**Table 1 | Baseline clinicopathological characteristics across all cohorts**

| Characteristics | Dataset, WSI No. | | | |
|---|---|---|---|---|
| | Training/Validation Cohort (*n* = 2544) | Test Cohort (*n* = 636) | External Test Cohort (*n* = 477) | *p*-value |
| Age at diagnosis, median (range), y | 58 (18–94) | 62 (20–92) | 62 (20–98) | 0.79 (ANOVA) |
| Sex (%) | | | | |
| Female | 53.2 | 50 | 52.8 | 0.71 (Chi$^2$) |
| Male | 46.8 | 50 | 47.2 | |
| Breslow Depth, median (range), mm | 0.94 (0.1–8.8) | 0.8 (0.12–8.0) | 1.3 (0.1–11.7) | 0.096 (ANOVA) |
| Miotic Count, median (range), mitoses/mm$^2$ | 1 (0–35) | 0.5 (0–13) | 0.5 (0–21) | 0.66 (ANOVA) |
| Anatomical Site (%) | | | | |
| Head and Neck | 41.9 | 44.3 | 24.5 | 4.88 × 10$^{-15}$ (Chi$^2$) |
| Upper Limb | 13.1 | 12.7 | 25.2 | |
| Lower Limb | 22.2 | 16.5 | 18.9 | |
| Trunk | 22.9 | 26.4 | 31.4 | |
| Histological Subtype (%) | | | | |
| Superficial Spreading Melanoma | 72.3 | 69.8 | 74.9 | 6.79 × 10$^{-23}$ (Chi$^2$) |
| Nodular Melanoma | 20.5 | 20.8 | 25.1 | |
| Lentigo Maligna Melanoma | 7.2 | 9.4 | 0 | |
| Metastasis (%) | | | | |
| Present | 10.4 | 10.4 | 16.4 | 0.083 (Chi$^2$) |
| Absent | 89.6 | 89.6 | 83.6 | |
| Time to Metastasis, median (range), m | 83 (1–267) | 87 (1–238) | 87 (3–234) | 0.22 (ANOVA) |

All *p*-values were derived using two-sided tests (Chi-squared for categorical variables, one-way ANOVA for continuous variables A *p*-value < 0.05 as considered statistically significant.
*WSI* whole slide image, *ANOVA* analysis of variance, *y* years, *mm* millimetres, *m* months.

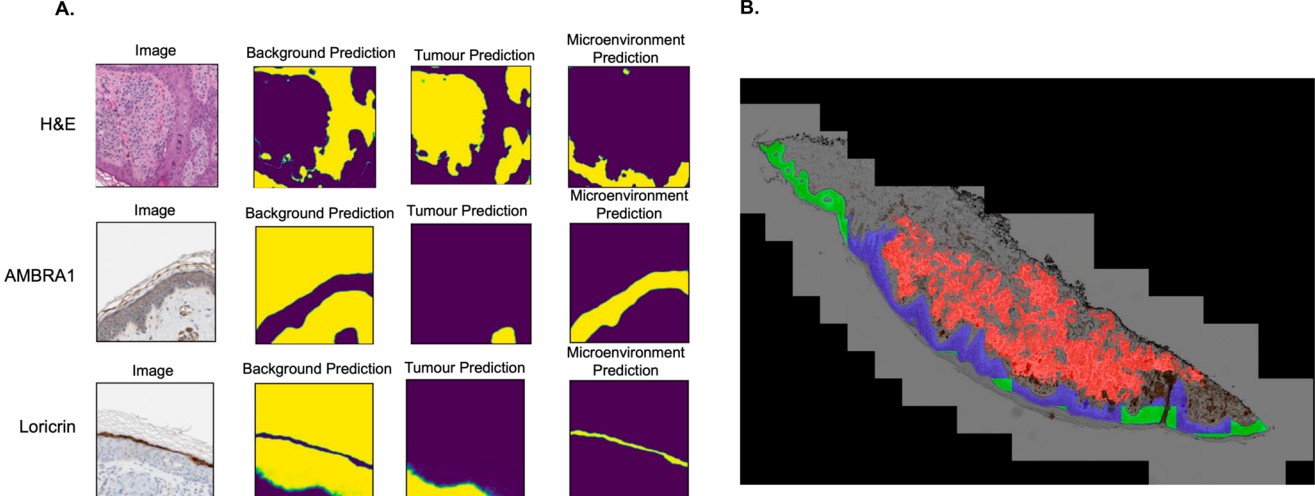

**Fig. 2 | Representative images of histological image analysis. A** Tile based predictions of segmentation in H&E, AMBRA1 and Loricrin stained whole slide images (WSI). The model segments the images into background, tumour regions, and tumour microenvironment. Yellow = Regions of Interest, Blue = Region of Non-Interest. **B** WSI based feature extraction using segmented tiles. Red = melanoma, blue = tumour microenvironment, green = extratumoural environment. This figure demonstrates the model's ability to distinguish tumour, microenvironment, and extratumoural regions, supporting the identification of prognostic features for melanoma risk stratification.

Following segmentation, feature extraction was performed using both deep learning-based analyses and handcrafted (HC) features. Differences in colour intensity and gradient distribution of the cells in the microenvironment were quantified using HC features. Cell spatial density and relation were quantified using a convolutional neural network (CNN) model based on the EfficientNet architecture. These analyses revealed several key TME-derived digital biomarkers. Loss of colour intensity and gradient in the TME as well as gaps in keratinocytes overlying the tumour were identified as potential markers of poorer outcomes. The final imaging predictions for each

WSI integrated the results from both HC and deep learning-derived features (Fig. 2).

## Multimodal prognostic performance

To enhance predictive performance, the clinical features Breslow depth, mitotic count, patient age, histological subtype, anatomical location and sex were integrated with image-feature analysis in the development of multimodal AI survival models. Three survival models were developed to combine imaging and clinical feature predictions: Cox proportional hazards (Cox), random survival forest (RSF), and

**Table 2 | Comparative performance of Cox, RSF and DeepSurv models using different features**

|  | Cox | RSF | DeepSurv |
|---|---|---|---|
| Staging Features Only | 0.66 (0.62–0.69) | 0.66 (0.61–0.71) | 0.58 (0.52–0.64) |
| Clinicopathological Features Only | 0.79 (0.72–0.86) | 0.80 (0.72–0.87) | 0.63 (0.58–0.69) |
| Imaging Features Only | 0.70 (0.60–0.78) | 0.64 (0.54–0.75) | 0.59 (0.54–0.65) |
| Multimodal (Combined Clinicopathological and Imaging Features) | 0.79 (0.69–0.88) | 0.82 (0.74–0.89)[a] | 0.63 (0.58–0.68) |
| Multimodal (Excluding Imaging Features) [b] | 0.79 (0.72–0.87) | 0.80 (0.73–0.86) | 0.60 (0.54–0.66) |

Cox Cox proportionate hazard survival analysis, RSF random survival forest.
[a]Melanoma Multimodal AI Prognostication (MelanoMAP).
[b]Ablation Study.

**A.**

**B.**

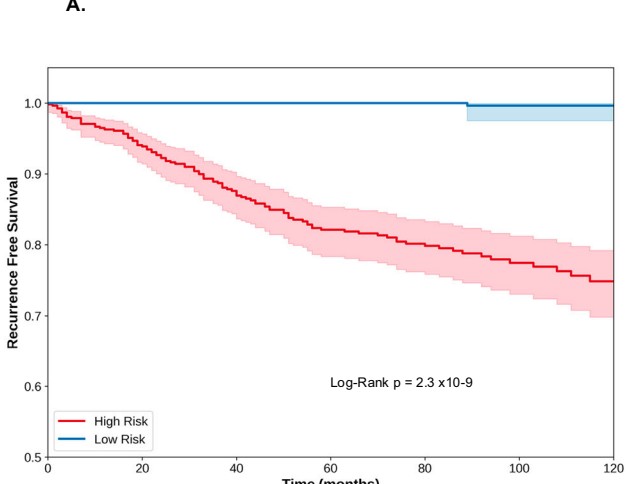
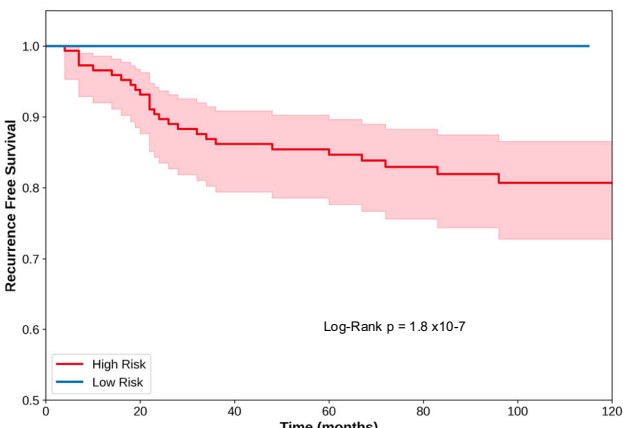

**Fig. 3 | Kaplan-Meier survival analysis to demonstrate MelanoMAP stratified patients into high-risk and low-risk groups. A** Survival curves for the test cohort incorporating data from the multinational database. **B** Survival curves for the external test cohort incorporating data from the Northern Ireland tissue biobank database, assessing model generalisability. The center line indicates the Kaplan-Meier estimate of survival probability; shaded areas represent 95% confidence intervals. The log-rank test ($p < 0.001$) confirms a statistically significant difference in recurrence-free survival between risk groups.

deep survival models (DeepSurv). Models were evaluated using clinicopathological-only, imaging-only, and combined feature sets. Bootstrapping with 1000 resamples on the external test was performed to generate 95% confidence intervals for each model. In the independent external test cohort, the combined RSF Model (MelanoMAP) achieved the highest discrimination performance with a C-index of 0.82 (CI 0.74–0.89) indicating robust predictive accuracy (Table 2). The combined multimodal model demonstrated an incremental performance improvement compared to clinicopathological-only (C-index 0.80, CI 0.73–0.87) and a marked improvement over imaging-only models (C-index 0.64, CI 0.54–0.75) (Table 2). Bootstrap-Based Paired Comparison did not reach significance, demonstrating a mean difference in C-index 0.004, 95% CI: (−0.046, 0.062). Most importantly, MelanoMAP significantly outperformed features of AJCC 8th edition staging, achieving a C-index of 0.82 (CI 0.74–0.89) compared to 0.66 (CI 0.61–0.71, $p < 0.001$).

Calibration curves are critical for evaluating the agreement between predicted and observed outcomes, ensuring that the model's survival predictions are not only accurate but also reliable across different risk groups. In addition to the C-index (0.82), calibration curves at 24 and 60 months demonstrated strong concordance between predicted and observed survival outcomes for clinicopathological-only or imaging-only features. Time-dependent Brier scores at 12, 24, 36, and 60 months further supported superior calibration of MelanoMAP, which consistently achieved the lowest score across all time points (Supplementary Table 2). To summarise model calibration across the entire time period, Integrated Brier Score (IBS) for each model was performed. The combined RSF model (MelanoMAP) achieved an IBS of

0.084, compared to 0.099 for the clinicopathological-only RSF model. Using 1000 bootstrapped resamples of the training data, the mean difference in IBS was 0.015 (95% CI: 0.007–0.028, $p = 0.032$), demonstrating a statistically significant improvement in calibration associated with the inclusion of imaging-derived features. These findings underscore the advantage of multimodal models like MelanoMAP in enhancing both discrimination and calibration performance for predicting survival outcomes melanoma patients.

### Risk stratification and clinical utility
Kaplan-Meier survival curves stratified patients into high-risk and low-risk groups based on the MelanoMAP model's risk assessment, using a threshold of 0.95 derived from the median predicted survival probability at 60 months, resulting in 50% of patients classified as low risk and 50% as high risk. Patients identified as high-risk exhibited a significantly lower recurrence-free survival probability, with a 5-year recurrence-free survival rate of approximately 81% (95% CI: 79–83%) compared to 98% (95% CI: 97–100%) in the low-risk group (Fig. 3). Log-rank test confirmed a highly significant difference between the two groups ($p < 0.001$) demonstrating clear separation of survival trajectories. In the independent test cohort, the model's performance was consistent, with high-risk patients showing a survival probability of approximately 84% (95% CI: 81–87%) at 5 years, compared to 100% survival in the low-risk group (Fig. 3). This process was replicated for the clinicopathological-only model to clarify the contribution of imaging features in the multimodal model (Supplementary Fig. 1). This demonstrated strong prognostic discrimination between high and low risk groups (log-rank $p < 0.001$). The MelanoMAP and

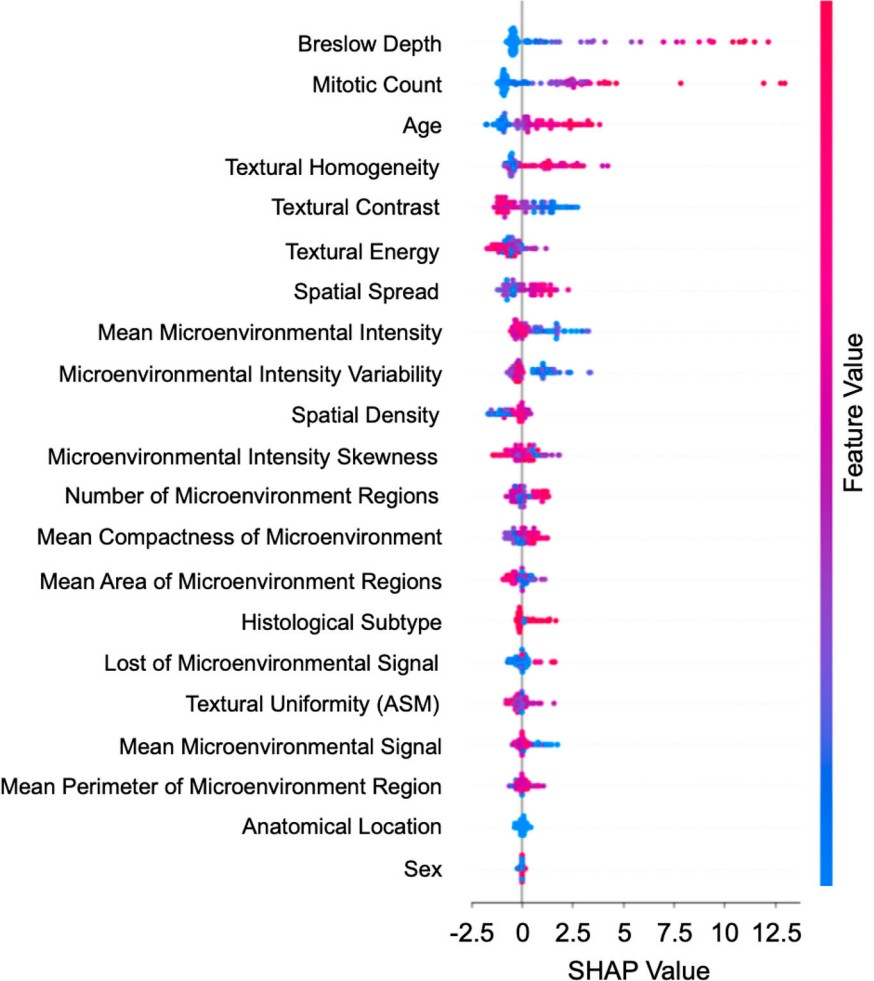

**Fig. 4 | Clinicopathological and microenvironmental features were critical for MelanoMAP performance.** SHAP (Shapley Additive Explanations) values illustrating the relative contribution of clinicopathological and microenvironmental features in predicting recurrence-free survival (RFS). Higher SHAP values indicate greater impact on model predictions, with a scale ranging from −2.5 to 12.5. Feature values are colour-coded (red = high values, blue = low values).

clinicopathological-only KM curves are comparable with slightly earlier divergence, more stable low-risk curve and slightly narrower confidence intervals in the MelanoMAP group. Collectively, these results validate the model's ability to generalise risk stratification and highlight its clinical utility for identifying patients with divergent survival outcomes and support the hypothesis that image derived features from the tumour microenvironment contribute additional prognostic information, particularly in identifying low-risk patients.

SHapley Additive exPlanations (SHAP) values were used to identify the most influential features of MelanoMAP in predicting metastasis risk, aligning these findings with established clinical domain knowledge (Fig. 4). Clinical features had the greatest impact on model performance. Higher mitotic counts were strongly associated with decreased survival probability, with the effect most pronounced at counts exceeding 3/mm². Tumours with a Breslow depth greater than 2 mm were linked to an increased likelihood of metastasis, while patients aged over 70 years faced a significantly elevated metastasis risk. Imaging features also provided strong prognostic signals. Increased texture contrast and reduced homogeneity were interpreted as markers of spatial heterogeneity and stromal compartmental irregularity, corresponding to neoangiogenesis, patchy inflammatory infiltrates, tumour budding, infiltrative growth patterns, and regressive fibrosis. Increased morphological density and microenvironmental area reflected stromal compression, cellular clustering, and expansion of the tumour stromal interface, associated with collagen deposition,

cohesive but irregular invasive fronts, and a broader spatial extent of microenvironmental involvement. Notably, microenvironmental area quantified the distribution of change across the WSI rather than its severity. These findings underscore the prognostic relevance of TME derived digital biomarkers and offer complementary biological insights beyond conventional clinical variables.

Finally, a Decision Curve Analysis (DCA) demonstrated that the Combined RSF Model provided the highest net benefit at risk thresholds below 10% (Fig. 5). This indicates that the model is particularly effective for identifying patients with a low predicted probability of metastasis, where closer follow-up or early intervention may still be warranted. Such findings highlight the model's potential utility in guiding localised CM management.

## Discussion

Prognostication of CM remains challenging, particularly for patients with localised disease, who account for most melanoma-related deaths[1-3]. Accurate prognostication is essential for guiding treatment and surveillance, yet current staging systems fail to capture the nuanced potential for disease progression, omitting key clinicopathological and microenvironmental factors[14,15]. Subtle changes within the TME, which play a critical role in tumorigenesis and metastasis, are often undetectable through conventional histopathological analysis[16]. This study addresses these gaps by integrating deep learning-derived imaging features with clinical data to develop a

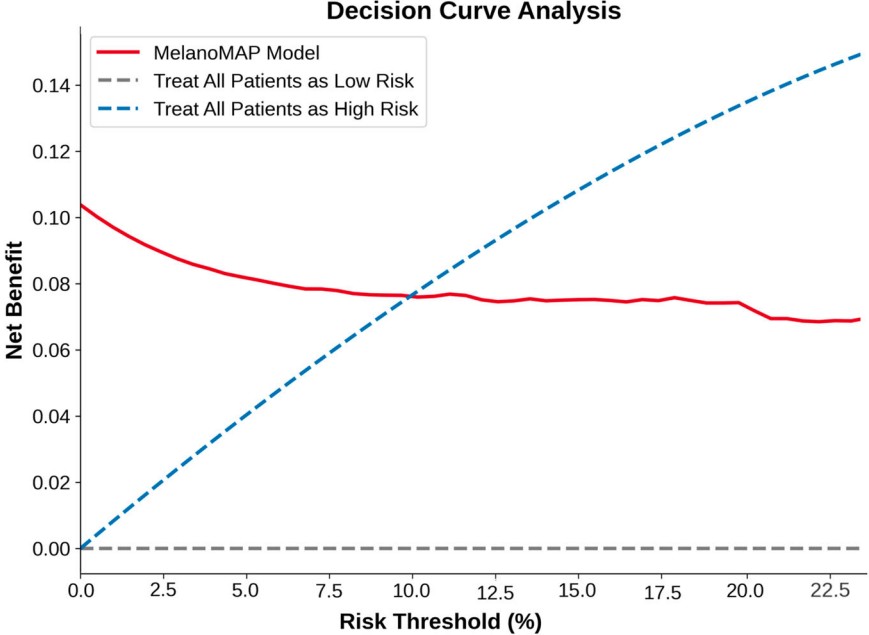

**Fig. 5 | Decision curve analysis (DCA) assessing the net benefit of MelanoMAP.** DCA compares the net benefit of the MelanoMAP model (red line) against two baseline strategies: treat all patients as low risk (gray dashed line) and treat all patients as high risk (blue dashed line). The analysis demonstrates that MelanoMAP provides the highest net benefit at risk thresholds below 10%, aligning with current clinical guidelines for melanoma prognostication.

multimodal AI framework, MelanoMAP. This model improves prognostication by incorporating histological, clinical, and tumour microenvironmental features, laying the groundwork for future biomarker discovery and AI-driven risk stratification.

This digital image analysis pipeline effectively captured prognostic features from histopathological slides, demonstrating consistent segmentation and feature extraction accuracy. Previous work by Kulkarni et al. demonstrated the potential of deep neural networks for predicting distant metastasis recurrence in melanoma histology, achieved AUCs of 0.905 and 0.88 on validation datasets[17]. However, their model focused on intrinsic tumour information overlooking the additional prognostic value offered within the TME[17]. Similarly, Comes et al. highlighted the prognostic utility of deep learning-based image analysis for disease-free survival in melanoma[18]. Clarke et al., in a meta-analysis, reported similar strong performance for automated image analysis in melanoma histological images[19]. While these studies report high performance metrics, their models were developed and tested on limited, homogeneous datasets, often from single institutions, which may have compromised generalisability due to overfitting. In contrast, our study utilised data from six institutions across five countries, three continents and included an external holdout test cohort to ensure robustness. By validating performance across diverse international datasets, our approach overcomes the limitations of single-institution studies, as highlighted by Combalia et al. [20,21] and enhances suitability for real-world clinical application. By incorporating TME-derived features and validating performance across diverse international datasets, our approach addresses the limitations of previous tumour-focused models and enhances its potential for broader clinical applicability in melanoma prognostication.

Survival models, including Cox proportional hazards, random survival forest (RSF), and DeepSurv, were employed to predict time-to-event outcomes. Unlike traditional classifiers, these models inherently account for censoring, providing insights into event timing. Our multimodal approach aligns with studies like Brinker et al.,[22] who combined imaging and clinical features to predict sentinel lymph node status, achieving notable performance with an AUROC of 61.8% for sentinel lymph node status classification. In contrast, MelanoMAP

achieved consistent performance across bootstrapped iterations, suggesting a reproducible, if incremental improvement over clinicopathological-only models highlighting the effectiveness of leveraging both clinicopathological and image derived features of the TME for CM prognostication. Most importantly, MelanoMAP significantly outperformed features of AJCC 8th edition staging, achieving a 24% improvement in discrimination accuracy for survival prediction. These findings underscore the potential of multimodal, data-driven prognostic models to enhance personalised risk stratification beyond conventional staging frameworks.

This study demonstrated a clear separation of survival curves in risk-stratified cohorts, underscoring the model's clinical applicability. It enables the identification of high-risk patients who may benefit from intensive therapies and closer follow-up, while reducing the likelihood of overtreatment for lower-risk patients. Importantly, the model demonstrated strong generalisability across independent cohorts, a key step toward real-world clinical implementation. MelanoMAP KM curves demonstrated earlier divergence, more stable low-risk curve and slightly narrower confidence intervals compared to other models supporting the hypothesis that image derived features from the tumour microenvironment contribute additional prognostic information, particularly in identifying low-risk patients. Our results align with recent advancements in survival prediction models leveraging deep learning. Notably, Kurz et al. utilised histopathological imaging in colon cancer to distinguish survival probabilities across risk groups, demonstrating the potential of such models to provide clinically actionable insights[23]. Similarly, MM-SurvNet combined histopathological and clinical data to achieve similar performance in breast cancer survival prediction, underscoring the value of multimodal approaches[13]. The improved stratification of the low-risk group in our model reflects findings from mechanistic IHC studies implicating AMBRA1 signalling a biomarker in low-risk CM[24]. These studies highlight the potential of integrating additional data modalities, such as genomic markers, to refine risk predictions further in the future.

SHAP analysis provided key insights into the factors contributing to metastasis risk, emphasising the potential for explainable AI in

augmenting clinical decision-making. The identification of age, Breslow depth, mitotic count, and TME as critical determinants aligns with established prognostic markers, while also highlighting imaging-derived features that go beyond traditional methods.

Age was identified as a significant risk factor, with patients aged 70 years or older showing a strong association with increased metastasis risk, as indicated by positive SHAP values. This observation is consistent with findings in the melanoma literature, where older age is frequently associated with worse outcomes due to age-related decline in immune function[25]. This highlights the need for tailored surveillance and consideration of adjuvant therapies in elderly patients, as such measures could mitigate the higher metastasis risk in this group.

Breslow depth was another highly important feature, with lesions exceeding 2 mm showing substantially higher SHAP values. This reinforces long-established evidence correlating tumour thickness with poor outcomes and align with current AJCC staging[26,27]. Our findings also emphasise the necessity for prolonged follow-up in localised thick melanomas as these patients exhibit additional risk beyond the routine follow-up period. This highlights the need for further studies to evaluate whether prolonged or more frequent surveillance in such cases, potentially extending beyond 5 years, could improve early detection of recurrences and patient outcomes.

The mitotic count also emerged as a critical feature, with rates exceeding 3/mm² contributing significantly to metastasis risk. This finding is particularly interesting as miotic count has been removed from the most recent AJCC staging system, mainly due to difficulty in consistent evaluation by histopathologists[4]. Despite its exclusion from the current staging, our findings align with several studies identifying mitotic activity as an independent prognostic factor in melanoma[28]. Highly proliferative tumours warrant more aggressive interventions. Based on these findings consideration of expanding parameters, including mitotic count into future melanoma staging systems should be examined.

TME imaging features, derived through deep learning and handcrafted features, provided biologically grounded insights into melanoma prognostication. High texture contrast and low homogeneity were associated with increased metastasis risk, reflecting spatial heterogeneity and stromal compartmental irregularity. These features correspond histologically to neoangiogenesis with aberrant vascular proliferation, non-brisk or patchy inflammatory infiltrates, loss of cohesive tumour architecture, including tumour budding and infiltrative growth patterns, and regressive stromal fibrosis[29]. Morphological features, such as increased regional density and expanded microenvironmental area, reflected stromal compression, cellular clustering, and greater spatial involvement of the tumour stroma interface. These patterns align with collagen deposition, disruption of the stromal scaffold, and altered cohesiveness at the invasive front, features increasingly recognised as contributors to metastatic potential[30]. Together, these findings support emerging evidence that digital representations of the TME can capture clinically relevant aspects of immune evasion and tissue remodelling[29,30]. The application of SHAP-based explainability further enhanced interpretability by linking imaging-derived features to established biological mechanisms, enabling transparent integration of deep learning outputs into clinical prognostic workflows[31].

The calibration and reliability of predictive models are critical for their integration into clinical workflows. In this study, MelanoMAP demonstrated strong agreement between predicted and observed survival probabilities, as reflected in calibration curves at 24 and 60 months as well as a significant improvement in Integrated Brier Scores across 12, 24, 36, and 60-month time points in MelanoMAP compared to all other models. This indicates a statistically significant improvement in calibration associated with this multimodal approach. These findings align with those of Carse et al., who emphasised the importance of post-hoc calibration in ensuring reliable predictions in

medical imaging[32]. Decision Curve Analysis (DCA) further validated the model's clinical utility, with the highest net benefit at risk thresholds below 10%. Interestingly, this threshold aligns with current clinical guidelines recommending sentinel lymph node biopsy (SLNB) for patients with a metastasis risk exceeding 10%[33,34]. MelanoMAP supports nuanced SLNB decisions by providing actionable insights, particularly for borderline cases. Taken together with the C-index results, these findings suggest that while clinical features remain the dominant predictors, TME-derived imaging features contribute measurable added value, most notably in refining risk estimates for low-risk patients, in line with prior IHC mechanistic work on TME signalling[24,35]. Although they do not substantially reclassify individuals into different risk groups, their inclusion improves calibration, enhancing the precision and reliability of predicted outcomes. This refinement in discriminative and calibration performance may hold meaningful clinical value for patient counselling and tailoring surveillance intensity.

Ulceration is a key clinical feature in melanoma prognostication. However, it was excluded from our cohorts to avoid introducing bias into the digital image analysis, as ulceration transforms the microenvironment by disrupting its structural integrity and introducing artefacts. These biases compromised feature extraction and model interpretation in pilot work, leading to their exclusion. Future work incorporating ulceration as a feature, while mitigating its confounding effects, could further enhance the model's ability to predict time to metastasis. This study demonstrated the advantage of incorporating IHC stains, including AMBRA1 and Loricrin, to capture microenvironmental changes in key cellular signalling pathways, particularly autophagy and differentiation, thereby supporting functional, biology-guided model explainability[36,37]. These stains provided crucial insights into the tumour microenvironment that may not be fully detectable on H&E-stained slides alone. However, having validated this approach, our findings support progression toward applying MelanoMAP to H&E-stained slides independently. This would extend the model's applicability to scenarios where access to IHC staining is limited, further establishing its universal potential for clinical use. Finally, information on molecular profiling such as BRAF mutation was not available in these cohorts. This information could improve the model performance and would support further tailored treatment recommendations such as the use of immune checkpoint inhibitors.

By integrating deep learning-based pathology image features with clinical data from international cohorts, MelanoMAP advances the prognostic landscape for localised melanoma addressing limitations of traditional clinical predictors included in the 8th edition of AJCC staging. This multimodal AI model reveals TME-derived digital biomarkers and improves prognostication by accurately predicting time to metastasis. By identifying patients within actionable risk thresholds, it supports precision oncology practices and may guide informed decisions regarding SLNB and adjuvant or neoadjuvant treatment.

## Methods

This study adhered to the Declaration of Helsinki guidelines and complies with all relevant ethical regulation which were obtained through the Newcastle University Dermatology Biobank (REC REF 24/NE/0014). The dataset consisted of 3657 whole slide images (WSIs) stained with haematoxylin and eosin (H&E), AMBRA1, and Loricrin, corresponding to 1219 patients with histologically confirmed CM. Data were collected from patients diagnosed with primary non-ulcerated AJCC (8th edition) stage I/II melanoma between 2004 and 2014, with a minimum of 10 years of clinical follow-up. The images were sourced from six international centers: University Hospital North Durham, Durham, UK; James Cook University Hospital, Middlesborough, UK; Roswell Park Comprehensive Cancer Center, Buffalo, USA; Peter MacCallum Cancer Centre, Melbourne, Australia; Hospital Clinic Barcelona, Barcelona, Spain; The Northern Ireland Tissue Biobank, Belfast, UK.

Patients aged ≥18 years and diagnostic biopsies with complete peripheral margins were included. Exclusion criteria encompassed ulcerated melanoma, acral lentiginous melanoma, non-cutaneous or mucosal melanoma, a history of previous melanoma, atypical mole syndrome, or multiple in-situ melanomas. Patients with unresectable stage III/IV melanoma, pregnancy, or inability to provide informed consent were also excluded. The primary outcome was Recurrence-Free Survival (RFS), defined as the time from complete excision to first recurrence.

## Preprocessing and feature selection

Key clinicopathological variables, including patient age, Breslow depth, and mitotic count, were standardised to mitigate biases and scale discrepancies. Sex was determined by electronic health records and not self-reported; gender identity was not recorded, and no sex-based subgroup analyses were performed. Collinearity checks excluded features with correlations exceeding ±0.7 (Supplementary Table 3). After these checks, the following features were retained for analysis: patient age, sex, anatomical location, histological subtype, Breslow depth, miotic count, mean microenvironmental signal, loss of microenvironmental Signal, mean microenvironmental intensity, microenvironmental intensity skewness, microenvironmental intensity variability, mean area of microenvironmental regions, mean compactness of microenvironment, mean perimeter of microenvironmental regions, number of microenvironmental regions, spatial spread, spatial density, textural uniformity (singular second moment), textural homogeneity, textural contrast, and textural energy.

## Data imputation

Three imputation methods were compared to address missing clinical data: mean imputation, k-nearest neighbours (KNN) imputation, and iterative random forest (RF)-based imputation. The iterative RF-based approach achieved the highest accuracy (87.3%). This method was applied solely to the training dataset, ensuring a complete dataset for model training. The validation and test datasets were complete with no missing data and, therefore, were not imputed to preserve data integrity and prevent leakage.

## Digital biomarker pipeline development

A total of 3,657 whole slide images (WSIs) stained with H&E, AMBRA1, and Loricrin were annotated by a team of four board-certified UK consultant dermatopathologists across three academic institutions. Immunostaining was performed at a UKAS-accredited (laboratory (Royal Victoria Infirmary, Newcastle upon Tyne Hospitals NHS Foundation Trust, UK). To minimise inter-laboratory variability, all slides were stained using the UKCA-marked AMBLor® diagnostic kit (AMLo Biosciences, UK), following harmonised protocols[35]. Following immunostaining, each WSI was annotated by two dermatopathologists. Annotation consensus meetings were then held, and group level concordance was achieved in >98% of cases. WSIs without consensus were excluded. Where slides contained multiple tumour regions, sections, or serial cuts, all visible tissue was annotated.

Annotated slides were processed using a deep learning pipeline designed to mirror histopathologist workflows. Tissue segmentation was performed using a modified U-Net architecture incorporating instance normalisation and a softmax output layer. The model was trained using a weighted cross-entropy loss to address class imbalance and achieved high segmentation performance with F1-scores of 0.92 (H&E), 0.95 (AMBRA1), and 0.94 (Loricrin), and mean Intersection-over-Union exceeding 0.90. The U-Net model segmented each WSI into tumour, epidermis, background, and immunostain regions. From these segmented maps, 256 × 256 pixel patches were extracted with 50% overlap, yielding ~3.2 million image patches. Sensitivity analyses comparing overlapping and non-overlapping patch sampling demonstrated that model performance was robust to patch extraction strategy.

Feature extraction followed a hybrid approach, combining handcrafted (HC) and deep learning-derived features. HC features included morphological and textural descriptors such as micro-environmental area, regional density, compactness, texture homogeneity, contrast, and skewness. Deep features were extracted from the penultimate layer of an EfficientNet-B0 model pretrained on ImageNet and fine-tuned on the histology patches, resulting in a 1280-dimensional embedding per patch. Patch level features were aggregated using mean pooling to create slide level representations, which were concatenated with clinicopathological variables and input to multimodal survival prediction models.

TME derived features were extracted specifically from non-tumour stromal and epidermal regions identified by U-Net segmentation. These features reflected spatial and textural variation within the TME, independent of tumour regions. For AMBRA1-stained slides, tumour microenvironmental staining intensity and gradient distribution were quantified, and IHC expression was contextualised relative to the normal epidermis, which served as an internal control. For Loricrin, CNN-based analysis identified expression gaps >20 microns, indicative of protein loss. All models were trained and validated on a per patient basis to prevent data leakage.

## Model development and evaluation

Survival models were developed to predict time-to-event outcomes using three approaches: Cox Proportional Hazards (Cox), Random Survival Forest (RSF), and Deep Survival Models (DeepSurv). These models were evaluated using a staging-only, clinicopathological-only, imaging-only, multimodal combined, and ablation study (multimodal excluding imaging) feature sets. The ablation study was performed to enable quanitification of the added value of imaging features in the metastasis prediction of the MelanoMAP. The performance of each model was assessed on the independent external Northern Ireland Tissue Biobank Database. Bootstrapping with 1000 resamples on the external test was performed to generate 95% confidence intervals for each model and bootstrap-based paired comparison was used to test statistical significance. Discrimination was measured using the concordance index (C-index), and calibration was evaluated with time-dependent and integrated Brier scores and calibration curves at 2–5 years.

To assess the clinical utility of the predictive models, decision curve analysis (DCA) was performed. DCA evaluates the net benefit of using a predictive model across a range of decision thresholds, allowing for comparison between the model, treating all patients, or treating none. For each threshold probability, patients were classified as high-risk if their predicted survival probability fell below 1–p and the net benefit calculated. Threshold probabilities were chosen within a clinically relevant range. The net benefit curves for each model, "treat all" strategy, and "treat none" strategy were plotted for comparison.

MelanoMAP generates a numerical risk score (0-1). Kaplan-Meier survival curves stratified patients into high-risk (MelanoMAP risk score ≥ 0.95) and low-risk (MelanoMAP risk score <0.95) groups. The threshold of 0.95 was derived from the median predicted survival probability at 60 months, resulting in approximately 50% of patients classified as low risk and 50% as high risk. This threshold was validated in an independent cohort, optimised sensitivity and specificity for metastasis prediction. SHAP analysis was employed to interpret feature importance within the MelanoMAP model. All packages, software and versions used in the study are listed in Supplementary Table 4.

## Reporting summary

Further information on research design is available in the Nature Portfolio Reporting Summary linked to this article.

# Data availability

Clinical and histopathological datasets from multiple institutions are under controlled access due to patient privacy, ethics, and institutional

agreements. Raw data cannot be deposited publicly. De-identified derived datasets (digital biomarkers, model inputs/outputs, data dictionary) are available upon request to Dr Tom Andrew (tom.andrew@newcastle.ac.uk) for non-commercial academic use, subject to a data use agreement and institutional approval. Requests reviewed within 4–6 weeks; data available for at least 5 years post-publication. Separate Source Data files are provided for each figure/table as individual Excel sheets. Source data are provided with this paper.

## Code availability

Data processing and analyses were conducted using Python programming language. All code used for data preprocessing, feature extraction, model training, and SHAP-based interpretation is publicly available at https://github.com/tomwandrew/MelanoMAP under the MIT License.

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

## Acknowledgements

This research was supported by a Clinical Fellowship grant to TWA from Cancer Research UK (CRUK) and a research grant from the North Eastern Skin Research Fund. PEL is the Chief Scientific Officer and PS is the Chief Histopathologist for AMLo Biosciences Ltd. MC received support through consultancy with AMLo Biosciences Ltd. For tissue samples from Melanoma Research Victoria (MRV), Australia, the authors acknowledge the MRV lead investigators Grant McArthur, Victoria Mar, Damien Kee and Craig Underhill. Samples received from the Northern Ireland Tissue Bank (NITB) in Belfast, UK, were supported by funds the NITB received from the HSC Research and Development Division of the Public Health Agency in Northern Ireland.

## Author contributions

T.A. contributed to the conceptualisation and design of this study. T.A., M.C. primarily handled and analysed the research data. All authors interpreted the results. T.A., M.C., and C.H. constructed machine learning and deep learning models and conducted statistical analysis. T.A. wrote the original draft of the paper. P.L., S.G., G.P., S.P., G.M., G.R., P.S., R.P., and S.S. revised the paper. All authors have read and approved the final manuscript.

## Competing interests

The authors declare no competing interests.
