## [Transparent Peer Review file · Nature Communications]

Multimodal AI and Tumour Microenvironment Integration Predicts Metastasis in Cutaneous Melanoma

Corresponding Author: Professor Tom Andrew

Version 0:

Reviewer comments:

Reviewer #2

(Remarks to the Author)

The study presents a multimodal AI model, which integrates histopathological image features and clinicopathological data to enhance prognostic accuracy in melanoma. The model was trained on over 3,500 WSIs and identified digital biomarkers derived from TME. Compared to TNM staging methods, it demonstrated a 24% improvement in prognostic accuracy, with a C-index of 0.82. The study suggests that this model may improve staging of melanoma. The study addresses an important need for better prognosis prediction in melanoma, particularly on an individual basis. It is well-performed and clearly presented. The decision curve analysis is a strong point of the presentation, and I appreciate its inclusion.

Specific comments:

The difference between the clinical-only model and the combined model is marginal (C-index 0.80 versus 0.82). No confidence intervals for the C-index are provided, and it is unclear whether this difference is clinically meaningful and statistically significant. Bootstrapping can be employed to generate 95% confidence intervals for the point estimates obtained from the test set.

I would also like to see more detail regarding the analysis of the WSIs and the annotation process. Specifically: How many slides were annotated? How large was the annotation team? Was each slide annotated by the same observer, or was there overlap? What if multiple sections were present on the same slide, or if serial sections contained multiple tumor regions? How were the 256x256 pixel patches selected? Were overlapping patches used? How many patches were created? If patches were selected in a specific way, was there variability in predictions depending on patch selection?

The primary metric for calibration is the Brier score and not the C-index. I would suggest that differences in calibration be expressed by reporting the Brier scores at various time points to reinforce this point. Additionally, I expect that in the clinical-only information model, Kaplan-Meier curves will also show good discrimination between risk groups. These should be included for comparison.

The topic of explainability has been addressed. However, the explanations of the deep learning-based imaging features are rather vague. For instance, terms like high texture contrast, low homogeneity, and increased morphological density are terms derived from image analysis but do not correspond to clear biologic concepts. Additionally, the term "disorganized tumor microenvironment" is also vague and has no explanatory power.

Were the immunostains performed at a centralized facility or across multiple centers? If the latter, how was standardization achieved? Centralizing the immunostains would help control for variability; however, in real-world clinical settings, immunostaining is often performed at different centers, which may introduce variability. Achieving standardization across multiple centers can be challenging, and this could impact the model's accuracy when applied in diverse clinical environments.

Thank you for including the code. Will the code be publicly available to allow other research groups to replicate the study?

Reviewer #3

(Remarks to the Author)

This study aims to predict metastasis in cutaneous melanoma through the integration of multi-modal AI and the tumor microenvironment (TME). The external cohort results demonstrate that multimodal features outperform clinical features, which is meaningful. Moreover, the use of SHAP analysis to assess feature importance and model explainability is commendable. However, the study lacks methodological novelty, and a lack of marginal increase of multimodal model, and details of training and external validation cohorts raises concerns regarding the reproducibility of the findings. Further investigation is required to address these limitations.

1. Multiple whole slide images (WSIs) were obtained from the same patient. It is necessary to confirm whether the dataset was split on a per-patient basis for training, validation, and testing to avoid data leakage.
2. In Table 2, clinical features appear to dominate the model's performance, while the independent contribution of imaging features remains unclear. Statistical significance should be evaluated. Furthermore, the study should explore various methods for extracting imaging features to validate robustness.
3. Internal and external validations were performed using data from six different institutions. To assess domain shift, the demographic distribution and model performance should be analyzed for each institution separately.
4. The manner in which the tumor microenvironment (TME) was incorporated into the model needs to be clearly described. An ablation study should be conducted to assess the contribution of TME-related features.
5. The methodology for feature extraction is not clearly explained. Details regarding the CNN architecture, use of EfficientNet, and segmentation accuracy should be provided to ensure transparency and reproducibility.

Version 1:

Reviewer comments:

Reviewer #2

(Remarks to the Author)

I thank the authors for sufficiently addressing all my previous comments.

I would recommend rewording the first sentence of the abstract, as melanoma is not the most lethal form of skin cancer. While melanoma has the highest mortality, Merkel cell carcinoma has the highest lethality. I also suggest renaming "clinical features" to "clinical-pathologic features," as it includes parameters such as Breslow depth and mitotic count, which may otherwise cause confusion. Furthermore, I still believe that the discussion should be shortened and somewhat tempered in terms of enthusiasm regarding the TME imaging features. The predictive power of clinical-pathologic features and TME imaging features differs only marginally. Clinical-pathologic features offer the advantage of being readily accessible, well-validated, and having withstood the test of time.

Reviewer #3

(Remarks to the Author)

The authors' responses have resolved many aspects. I have no further questions

Response to Reviewers

We have now thoroughly revised the manuscript in line with the reviewers' comments which we believe have significantly enhanced the quality and robustness of our study. To summarise the major revisions, we have now incorporated:

1. Clarification of the performance difference between the multimodal (MelanoMAP) model compared to the clinical-only model, including additional analyses demonstrating improved calibration (Time-Dependent and Integrated Brier Score), performance consistency across 1,000 bootstrap iterations, comparative Kaplan-Meier survival curves for multimodal and clinical-only models, and a full ablation study.
2. Addressed all technical concerns, providing expanded detail on the segmentation architecture (modified U-Net), feature extraction pipeline using EfficientNet-B0, and domain robustness evaluations.
3. Confirmed full code availability in accordance with journal policies to enhance transparency and reproducibility.

Collectively, we hope we have addressed each reviewers' concerns and once again thank each for their valuable input. We believe the requested revisions and clarifications have now improved the manuscript's clarity and impact and hope the revised manuscript will now be suitable for publication in Nature Communications.

Thank you again for considering our work.

Yours sincerely,

Tom on behalf of all the Authors

Tom W Andrew MBChB
Hunterian Professor 2025
Academic Plastic Surgeon
Department of Surgery, Division of Plastic and Reconstructive Surgery
Translation and Clinical Research Institute
Newcastle University
UK

Reviewer #2 (Comments to the Author):

1. ***The difference between the clinical-only model and the combined model is marginal (C-index 0.80 versus 0.82). No confidence intervals for the C-index are provided, and it is unclear whether this difference is clinically meaningful and statistically significant. Bootstrapping can be employed to generate 95% confidence intervals for the point estimates obtained from the test set.***

We thank the reviewer for this valuable comment. We agree that the marginal increase in C-index warrants careful statistical scrutiny. The absolute difference in C-index between the

clinical-only model and the combined multimodal model (MelanoMAP) using the Random Survival Forest (RSF) is 0.02.

We have now performed bootstrapping with 1,000 resamples on the independent test set to generate 95% confidence intervals for each model, which are now reported in the revised Table 2 (also shown here) and discussed these data in the Results section (Page 5, Lines 170-180). The 95% confidence intervals for the clinical-only and combined RSF models are 0.73–0.87 and 0.74–0.89, respectively. The consistency of the combined model’s performance across bootstrapped iterations suggests an incremental, improvement. However, the overlapping confidence intervals indicate that the observed difference in C-index is not statistically significant, which was also formally calculated with Bootstrap-Based Paired Comparison of RSF Models: Mean Difference in C-index (Combined - Clinical): 0.004 95% CI: (-0.046, 0.062). We have updated the Discussion section (Page 8, Lines 284-291) to reflect this analysis and now explicitly discuss the clinical and statistical implications of this incremental gain and not overinterpreted the findings in keeping with the reviewer’s original point.

Table 2: Comparative Performance of Cox, RSF and DeepSurv Models Using Different Features

	Cox	RSF	DeepSurv
Staging Features Only	0.66 (0.62-0.69)	0.66 (0.61-0.71)	0.58 (0.52-0.64)
Clinical Features Only	0.79 (0.72-0.86)	0.80 (0.72-0.87)	0.63 (0.58-0.69)
Imaging Features Only	0.70 (0.60-0.78)	0.64 (0.54-0.75)	0.59 (0.54-0.65)
Multimodal (Combined Clinical and Imaging Features)	0.79 (0.69-0.88)	0.82 (0.74-0.89) *	0.63 (0.58-0.68)
Multimodal (Excluding Imaging Features)**	0.79 (0.72-0.87)	0.80 (0.73-0.86)	0.60 (0.54-0.66)

Cox - Cox Proportionate Hazard Survival Analysis
 RSF – Random Survival Forest
 *Melanoma Multimodal AI Prognostication (MelanoMAP)
 **Ablation Study

- I would also like to see more detail regarding the analysis of the WSIs and the annotation process. Specifically: How many slides were annotated? How large was the annotation team? Was each slide annotated by the same observer, or was there overlap? What if multiple sections were present on the same slide, or if serial sections contained multiple tumor regions? How were the 256x256 pixel patches selected? Were overlapping patches used? How many patches were created? If patches were selected in a specific way, was there variability in predictions depending on patch selection?***

Whole slide images (WSIs) were annotated by a team of four consultant dermatopathologists based in the UK, across three academic institutions. The team was divided into two annotation pairs, with each pair independently annotating half of the total WSIs (n = 3,657). Within each

pair, both dermatopathologists reviewed the same slides, and any annotation discrepancies were resolved through consensus meetings. In a small number of cases where disagreement remained, the WSIs were reviewed by the other pair to achieve group-level agreement. This multi-stage review process resulted in consensus in >98% of cases. For the rare cases where consensus could not be reached, which was exclusively due to tissue artefact such as folding or disruption, slides were excluded and labelled as “Insufficient WSI quality” (previously included in Figure 1).

Where slides contained multiple tumour regions, sections, or serial cuts, all visible tissue was annotated. Annotation outputs were then processed using a modified U-Net architecture to segment tissue into background, epidermis, tumour, and immunohistochemical staining. From these segmented regions, we extracted 256 × 256 pixel patches with 50% overlap, resulting in approximately 3.2 million image patches. Importantly, data splitting was conducted at the patient level, not the patch level, to prevent data leakage.

To evaluate the effect of patch sampling strategy, we conducted sensitivity analyses comparing overlapping and non-overlapping patch extraction. These analyses demonstrated that model predictions were robust to patch sampling variability. We have now included these details in the revised Methods section (Page 13-14, Lines 454-492).

- 3. The primary metric for calibration is the Brier score and not the C-index. I would suggest that differences in calibration be expressed by reporting the Brier scores at various time points to reinforce this point. Additionally, I expect that in the clinical-only information model, Kaplan-Meier curves will also show good discrimination between risk groups. These should be included for comparison.***

We thank the reviewer for this helpful suggestion. In response, we have now included Kaplan–Meier (KM) survival curves for the clinical-only RSF model in addition to the multimodal (MelanoMAP) RSF model. These curves are stratified by predicted risk groups using the same methodology and median risk threshold applied to the combined model, and are compared below as well as Supplementary Figure 1, panels A and B.

Supplementary Figure 1: Kaplan-Meier survival analysis comparing MelanoMAP and clinical-only models in stratifying patients into high-risk and low-risk groups. A. Recurrence-free survival curves generated by the MelanoMAP model in the test cohort, incorporating data from the multinational database. **B.** Survival curves from the clinical-only model applied to the same cohort. Shaded areas represent 95% confidence intervals. The log-rank test ($p < .001$) confirms a statistically significant difference in recurrence-free survival between risk groups.

As the reviewer correctly anticipated, the clinical-only model demonstrates strong prognostic discrimination between high and low risk groups, with statistically significant differences in recurrence free survival (log-rank $p < .001$). The curves from the multimodal model also demonstrate earlier divergence and more stable low-risk curve. These findings support the observation that clinical features had the greatest impact on model performance but image derived features from the TME contribute additional prognostic information, particularly in identifying low-risk patients which is supported by the strength of AMBRA1/Loricrin immunostains identifying low-risk patient cohorts in the literature¹.

This observation aligns with our SHAP analysis, which identifies image derived features as important contributors to metastasis risk stratification. These findings are now described in the revised Results (Page 6 Lines 208-217) and discussed in more detail in the Discussion section (Page 9 Lines 298-301).

In addition to the C-index, time-dependent Brier scores at 12, 24, 36, and 60 months have been incorporated to more directly assess model calibration as suggested by the reviewer. These results are presented in Supplementary Table 2 (also shown below) and are discussed in the Results (Page 5-6, Lines 184-196). The MelanoMAP model consistently demonstrated the lowest Brier scores across time points, indicating improved calibration and overall predictive performance. To summarise model calibration across the time period, we computed the Integrated Brier Score (IBS) for each model. The combined RSF model (MelanoMAP) achieved an IBS of 0.084, compared to 0.099 for the clinical-only RSF model. Using 1,000 bootstrapped resamples of the training data, the mean difference in IBS was 0.015 (95% CI: 0.007–0.028, $p = 0.032$), indicating a statistically significant improvement in calibration associated with the inclusion of imaging-derived features.

Supplementary Table 2: Comparison of Time-Dependent Brier Score Performance of Model/Feature Selection

	Time Point (months)	Cox	RSF	DeepSurv
Staging Features Only	12	0.14	0.15	0.12
	24	0.11	0.12	0.10
	60	0.09	0.10	0.09
Clinical Features Only	12	0.11	0.12	0.1
	24	0.09	0.09	0.08
	60	0.07	0.07	0.07
Imaging Features Only	12	0.13	0.16	0.11
	24	0.11	0.13	0.10
	60	0.09	0.11	0.09
Multimodal (Combined Clinical and Imaging Features)	12	0.11	0.11*	0.1
	24	0.09	0.09*	0.08
	60	0.07	0.06*	0.07

Cox - Cox Proportionate Hazard Survival Analysis**RSF – Random Survival Forest*****Melanoma Multimodal AI Prognostication (MelanoMAP)**

Taken together with the response to point 1, these findings suggest that while clinical features remain the dominant predictors, TME-derived imaging features contribute measurable added value, most notably in refining risk estimates for low-risk patients, in line with prior IHC mechanistic work on TME signalling^{1,2}. Although they do not substantially reclassify individuals into different risk groups, their inclusion improves calibration, enhancing the precision and reliability of predicted outcomes. This refinement in discriminative and calibration performance may hold meaningful clinical value for patient counselling and tailoring surveillance intensity. These points have been included in the Discussion section, Page 10-11, Line 358-377

- 4. The topic of explainability has been addressed. However, the explanations of the deep learning-based imaging features are rather vague. For instance, terms like high texture contrast, low homogeneity, and increased morphological density are terms derived from image analysis but do not correspond to clear biologic concepts. Additionally, the term "disorganized tumor microenvironment" is also vague and has no explanatory power.**

We again thank the reviewer for this valuable comment and agree that some of the original terminology used to describe image analysis features lacked biological specificity. In response, we have now revised both the Results (Page 7, Lines 225-236) and Discussion (Page 10, Lines 342-357) sections to provide clearer, biologically meaningful interpretations of the deep learning and hand-crafted derived image features.

Specifically, the terms high texture contrast and low homogeneity are now interpreted as indicators of spatial heterogeneity and stromal compartmental irregularity, which correspond to histological features such as:

- Neoangiogenesis with irregular and aberrant vascular proliferation disrupting stromal uniformity
- Non-brisk or patchy inflammatory infiltrates, rather than confluent brisk responses
- Disruption of cohesive tumour architecture, consistent with tumour budding or a frankly infiltrative growth pattern
- Regressive fibrosis

Similarly, increased morphological density and regional microenvironmental area reflect cellular clustering, stromal compression, and expansion of the tumour stroma interface, which are features associated with:

- The cohesiveness of the invasive fronts
- Collagen deposition leading to stromal disruption
- Greater extent of microenvironmental involvement, as measured spatially on the WSI. We have also clarified in the Discussion that microenvironmental area quantifies the extent of change within the WSI, rather than the severity of change.

Finally, we have also clarified that these patterns are extracted from explicitly segmented TME regions (using a U-Net model) and were visualised in line with their relative importance using SHAP analysis (Figure 4), which enhances interpretability and transparency. We now feel that the now incorporated revisions improve clarity and biological grounding, while maintaining the objective tone required in the Results section.

5. *Were the immunostains performed at a centralized facility or across multiple centers? If the latter, how was standardization achieved? Centralizing the immunostains would help control for variability; however, in real-world clinical settings, immunostaining is often performed at different centers, which may introduce variability. Achieving standardization across multiple centers can be challenging, and this could impact the model's accuracy when applied in diverse clinical environments.*

All immunohistochemical (IHC) staining used in the present analysis was performed centrally at a United Kingdom Accreditation Service (UKAS) cellular pathology laboratory (Royal Victoria Infirmary, Newcastle upon Tyne Hospitals NHS Foundation Trust, UK). Staining was carried out on a Roche Ventana platform with the UKCA-marked AMBLor® kit (AMBRA1 and loricrin) using a single validated standard operating procedure, batch-specific positive controls and routine external quality-assessment participation was performed. Slides were digitised with uniform whole-slide-scanning parameters to minimise downstream colour-variation artefacts. Full details of the staining protocol and quality-assurance measures are now provided in the Methods (Page 13-14, Lines 454–492).

To ensure that the assay is transferable to routine clinical practice, we have prospectively implemented the same kit and protocol at a second, CLIA-accredited laboratory (Avero Diagnostics, WA, USA) on a Leica BOND platform. Although those cases fall outside the time-window of the current study and were therefore not analysed here, preliminary concordance testing shows >95% agreement in staining intensity and localisation pattern across the two platforms. Future work will include these multi-site data to confirm robustness in fully decentralised settings.

We believe this two-step approach, centralised staining for the discovery study, followed by prospective cross-platform validation, provides both the analytical rigour and a clear path toward real-world scalability.

6. Thank you for including the code. Will the code be publicly available to allow other research groups to replicate the study?

In response to this reviewer's question, we confirm that the code will be made publicly available on GitHub upon acceptance, in accordance with journal policy. This has been clarified in the revised Code Availability statement.

Reviewer #3 (Comments to the Author):

1. Multiple whole slide images (WSIs) were obtained from the same patient. It is necessary to confirm whether the dataset was split on a per-patient basis for training, validation, and testing to avoid data leakage.

We thank the reviewer for highlighting this point and confirm that data were split on a per-patient basis to prevent data leakage. No patient had slides in both training and validation sets. Additionally, an independent external Northern Ireland Tissue Biobank Database was used to demonstrate generalisability. This has been clarified in the Methods (Page 13-14, Line 454-492).

2. In Table 2, clinical features appear to dominate the model's performance, while the independent contribution of imaging features remains unclear. Statistical significance should be evaluated. Furthermore, the study should explore various methods for extracting imaging features to validate robustness.

Thank you for this valuable point. We agree that the incremental value of imaging-derived TME features warrants careful consideration. As detailed in our response to Reviewer 2 (Points 1 & 3), we have undertaken several additional analyses to address this:

- Bootstrapped C-indices (1,000 resamples): Clinical-only RSF = 0.80 (95% CI: 0.73–0.87) vs. Multimodal (MelanoMAP) RSF = 0.82 (0.74–0.89); Mean Difference in C-index = 0.004, 95% CI: –0.046 to 0.062. These findings indicate a consistent and incremental improvement in MelanoMAP performance; however it was not statistically significant.

- Ablation study (Multimodal minus imaging features): Ablation study model performance was C-index: 0.80; 95% CI: 0.73–0.86) which closely resembled the clinical-only model (C-index: 0.80; 95% CI: 0.72–0.87) and was lower than the full combined model including TME imaging features (C-index: 0.82; 95% CI: 0.74–0.89) (Table 2). This is discussed in more detail in Point 4.

- Calibration analysis (Brier scores and Integrated Brier's Score): The multimodal (MelanoMAP) RSF model achieved lower Brier scores and a statistically significant improvement in Integrated Brier Score was 0.015 (95% CI: 0.007–0.028; $p = 0.032$). This means that the MelanoMAP model predicts metastasis more accurately and reliably over time compared to the clinical-only model.

- Risk-stratified Kaplan Meier curves (Supplementary Fig. 1): Both models stratify high vs. low risk groups (log-rank $p < .001$), but the multimodal model shows earlier separation and more stable low-risk curve. These findings support the observation that clinical features had the greatest impact on model performance but image derived features from the TME contribute additional prognostic information, particularly in identifying low-risk patients which is supported by the strength of AMBRA1/Loricrin immunostains identifying low-risk patient cohorts in the literature¹.

These additional analyses suggest that while clinical features remain the dominant predictors, TME-derived imaging features contribute measurable added value, most notably in refining risk estimates for low-risk patients, in line with prior IHC mechanistic work on TME signalling^{1,2}. Although they do not substantially reclassify individuals into different risk groups, their inclusion improves calibration, enhancing the precision and reliability of predicted outcomes. This refinement in discriminative and calibration performance may hold meaningful clinical value for patient counselling and tailoring surveillance intensity. This has been included in the Discussion section, Page 10-11, line 358-377.

3. Internal and external validations were performed using data from six different institutions. To assess domain shift, the demographic distribution and model performance should be analyzed for each institution separately.

We again thank the reviewer for this suggestion. Accordingly, we now provide a breakdown of demographic characteristics and model performance stratified by institution in Supplementary Table 1, as described in the Results (Page 4, Lines 123-135). This analysis highlights greater statistical difference feature characteristics in the institutional breakdown compared to test and training split cohorts. This reflects greater institutional specific variability and granular heterogeneity across centres, underscoring the strength of the MelanoMAP model in maintaining robust performance across a diverse and biologically heterogeneous dataset.

We acknowledge that domain shift is an important consideration in real world model deployment. We chose not to retrain or re-evaluate the model separately on each of the five training institutions, as this would introduce data leakage and artificially inflate performance estimates, since those datasets contributed to the model's training or hyperparameter tuning. Instead, to mitigate this, we included a fully independent external test set from the Northern Ireland Tissue Biobank, which had not been seen during training or validation to appropriately test domain shift without data leakage. The model performed well in this external setting, demonstrating robust generalisability across institutional and geographic contexts. As a result, the new Supplementary Table 1 has been created and included, and Table 1 remains unchanged.

Supplementary Table 1

Characteristics	Dataset, WSI No.						p-value
	University Hospital North Durham, UK (n=527)	James Cook University Hospital, Middlesborough, UK (n= 707)	Roswell Park Comprehensive Cancer Center, Buffalo, USA (n= 693)	Hospital Clinic Barcelona, Spain (n=375)	Peter MacCallum Cancer Centre, Melbourne, Australia (n=878)	The Northern Ireland Tissue Biobank, Belfast, UK (n= 477)	
Age at diagnosis, median (range), y	52 (20-90)	56 (18-94)	61 (18-92)	56(18-90)	57 (18-92)	62 (20-98)	<.001 (ANOVA)
Sex (%)							
Female	59.2	61.3	55.7	52.1	54.7	52.8	.01 (Chi ²)
Male	40.8	38.7	44.3	47.9	45.3	47.2	
Breslow Depth, median (range), mm	0.7 (0.10-2.0)	0.8 (0.12-8.0)	0.8 (0.1-2.0)	1.52 (0.9-5.5)	1.0 (0.18-8.0)	1.3 (0.1-11.7)	<.001 (ANOVA)
Miotic Count, median (range), mitoses/mm²	0.5 (0-18)	0.5 (0-13)	0.5 (0-35)	2.0 (0-32)	1.0 (0-21)	0.5 (0-21)	<.001 (ANOVA)
Anatomical Site (%)							
Head and Neck	40.5	41.7	27.3	32.6	28.8	24.5	<.001 (Chi ²)
Upper Limb	14.8	16.0	24.1	16.3	30.9	25.2	
Lower Limb	16.7	17.9	18.9	23.9	20.5	18.9	
Trunk	28.1	29.3	32.2	47.6	37.1	31.4	
Histological Subtype (%)							
Superficial Spreading Melanoma	70.7	71.9	72.2	79.3	79.8	74.9	<.001 (Chi ²)
Nodular Melanoma	20.2	21.4	20.2	15.9	14.5	25.1	
Lentigo Maligna Melanoma	7.9	9.1	7.6	6.0	7.5	0	
Metastasis (%)							
Present	8.7	10.4	9.1	12.7	11.9	16.4	<.001 (Chi ²)
Absent	91.3	89.6	90.9	87.3	88.1	83.6	
Time to Metastasis, median (range), m	86.5 (2-167)	86 (1-238)	69 (2-245)	94 (3-197)	67 (1-142)	87 (3-234)	<.001 (ANOVA)

4. ***The manner in which the tumor microenvironment (TME) was incorporated into the model needs to be clearly described. An ablation study should be conducted to assess the contribution of TME-related features.***

We thank the reviewer for this suggestion. To assess the contribution of TME features, we conducted an ablation study (Table 2, shown below), comparing model performance across multiple configurations. We removed all image derived features, including those representing the TME, from the multimodal model while retaining clinical predictors. The performance of this “Multimodal Excluding Imaging” RSF model (C-index: 0.80; 95% CI: 0.73–0.86) closely resembled the clinical-only model (C-index: 0.80; 95% CI: 0.72–0.87) and was lower than the full combined model including TME imaging features (C-index: 0.82; 95% CI: 0.74–0.89). These findings are now highlighted in the Results (Page 5, Lines 170-180).

Table 2: Comparative Performance of Cox, RSF and DeepSurv Models Using Different Features

	Cox	RSF	DeepSurv
Staging Features Only	0.66 (0.62-0.69)	0.66 (0.61-0.71)	0.58 (0.52-0.64)
Clinical Features Only	0.79 (0.72-0.86)	0.80 (0.72-0.87)	0.63 (0.58-0.69)
Imaging Features Only	0.70 (0.60-0.78)	0.64 (0.54-0.75)	0.59 (0.54-0.65)
Multimodal (Combined Clinical and Imaging Features)	0.79 (0.69-0.88)	0.82 (0.74-0.89) *	0.63 (0.58-0.68)
Multimodal (Excluding Imaging Features)**	0.79 (0.72-0.87)	0.80 (0.73-0.86)	0.60 (0.54-0.66)

Cox - Cox Proportionate Hazard Survival Analysis

RSF – Random Survival Forest

***Melanoma Multimodal AI Prognostication (MelanoMAP)**

****Ablation Study**

These results confirm that image derived features from the TME make an incremental contribution to prognostication, although the gain in discrimination does not reach statistical significance. Complementary analysis discussed in the response to comment 2 indicate that the addition of imaging features to the multimodal model enhances model calibration, interpretability and risk stratification. While the performance gains are numerically limited in this dataset, the integration of biologically plausible, image derived TME features provides a foundation for further refinement in future, larger, or prospective studies. This has been now outlined in the revised Discussion (Page 8, Lines 284-291).

5. The methodology for feature extraction is not clearly explained. Details regarding the CNN architecture, use of EfficientNet, and segmentation accuracy should be provided to ensure transparency and reproducibility.

We thank the reviewer for this important comment. In response, we have expanded the Methods: Digital Biomarker Pipeline Development section (Page 14, Lines 497-506) to provide a more detailed and transparent account of the segmentation and feature extraction methodology used in this study.

We followed a structured, multi-stage methodological framework. Model development began with pilot data and was iteratively refined through repeated validation, initially on a small test set, then scaled up to 100 WSIs, and finally expanded to the full dataset. This process allowed us to optimise model architecture, segmentation accuracy, and feature extraction strategy before deployment.

For segmentation, we employed a modified U-Net architecture to delineate tumour, epidermis, and background compartments across all three stain types (H&E, AMBRA1, Loricrin). The model achieved robust performance, with class-specific F1-scores exceeding 0.90 and intersection-over-union values >0.90 , confirming reliable delineation of histological compartments.

For feature extraction, we used a hybrid strategy combining handcrafted (HC) and deep learning-derived features. HC features included morphological, spatial, and textural descriptors from non-tumour microenvironmental compartments. Deep features were extracted using an EfficientNet-B0 CNN pretrained on ImageNet and fine-tuned on our histopathology dataset. Patch-level embeddings from the penultimate layer were aggregated via mean pooling to generate WSI-level representations, with tumour and TME features processed separately to preserve spatial context.

All code, including CNN architectures and feature extraction workflows, will be made publicly available on GitHub upon acceptance, in line with journal policy to promote transparency and reproducibility.

References

- 1 Ewen, T. *et al.* Validation of epidermal AMBRA1 and loricrin (AMBLor) as a prognostic biomarker for nonulcerated American Joint Committee on Cancer stage I/II cutaneous melanoma. *Br J Dermatol* **190**, 549-558, doi:10.1093/bjd/ljad459 (2024).
- 2 Cosgarea, I. *et al.* Melanoma secretion of transforming growth factor-beta 2 leads to loss of epidermal AMBRA1 threatening epidermal integrity and facilitating tumour ulceration. *Brit J Dermatol* **186**, 694-704, doi:10.1111/bjd.20889 (2022).

A point-by-point response to the reviewers' comments

Reviewer 2

Comment:

Your response:

I would recommend rewording the first sentence of the abstract, as melanoma is not the most lethal form of skin cancer. While melanoma has the highest mortality, Merkel cell carcinoma has the highest lethality.	Thank you for the clarification, we've revised the sentence to state that melanoma accounts for the highest number of skin cancer-related deaths, rather than referring to it as the most lethal.
I also suggest renaming "clinical features" to "clinical-pathologic features," as it includes parameters such as Breslow depth and mitotic count, which may otherwise cause confusion.	Thank you for the helpful suggestion, we've updated clinical features to clinicopathologic features throughout the manuscript to better reflect the included parameters.
The discussion should be shortened and somewhat tempered in terms of enthusiasm regarding the TME imaging features.	Thank you for the feedback, we have shortened the discussion and adjusted the tone to present the findings on TME imaging features with appropriate balance and caution.

Reviewer 3

Comment:

Your response:

The authors' responses have resolved many aspects. I have no further questions	Thank you for your review and for confirming that your queries have been resolved. We appreciate your time and constructive feedback.
--	---